# Near-Optimal Experiment Design in Linear non-Gaussian Cyclic Models

**Ehsan Sharifian**
EPFL, Lausanne, Switzerland
ehsan.sharifian@epfl.ch

**Saber Salehkaleybar**
Leiden University, Leiden, The Netherlands
s.salehkaleybar@liacs.leidenuniv.nl

**Negar Kiyavash**
EPFL, Lausanne, Switzerland
negar.kiyavash@epfl.ch

## Abstract

We study the problem of causal structure learning from a combination of observational and interventional data generated by a linear non-Gaussian structural equation model that might contain cycles. Recent results show that using mere observational data identifies the causal graph only up to a permutation-equivalence class. We obtain a combinatorial characterization of this class by showing that each graph in an equivalence class corresponds to a perfect matching in a bipartite graph. This bipartite representation allows us to analyze how interventions modify or constrain the matchings. Specifically, we show that each atomic intervention reveals one edge of the true matching and eliminates all incompatible causal graphs. Consequently, we formalize the optimal experiment design task as an adaptive stochastic optimization problem over the set of equivalence classes with a natural reward function that quantifies how many graphs are eliminated from the equivalence class by an intervention. We show that this reward function is adaptive submodular and provide a greedy policy with a provable near-optimal performance guarantee. A key technical challenge is to efficiently estimate the reward function without having to explicitly enumerate all the graphs in the equivalence class. We propose a sampling-based estimator using random matchings and analyze its bias and concentration behavior. Our simulation results show that performing a small number of interventions guided by our stochastic optimization framework recovers the true underlying causal structure.

## 1 Introduction

Learning causal relationships among variables of interest in a complex system is a central goal in empirical sciences, forming the foundation for prediction, intervention, and explanation [25]. These relationships are typically represented by causal graphs in which an edge from variable $X$ to variable $Y$ indicates that $X$ is a direct cause of $Y$. Much of the existing literature on causal structure learning assumes that the underlying causal graph is a Directed Acyclic Graph (DAG). However, many natural and engineered systems include feedback mechanisms that give rise to cycles in their causal representations. Such cyclic structures emerge in equilibrium models, low-frequency temporal sampling of dynamical systems, and various biological networks [3].

In acyclic settings, a variety of methods (such as those based on conditional independence tests) allow us to recover the skeleton and orientations of the causal graph from observational data. These methods often fail when applied to graphs with cycles. More specifically, observational data alone does not even suffice for learning the skeleton, let alone orienting the edges of the graph [23].

39th Conference on Neural Information Processing Systems (NeurIPS 2025).

Interventions, i.e., actively perturbing the system and observing the resulting distributional changes can allow us to learn the graphs even with cycles. The *experiment design* problem studies how to best design interventions in order to maximize the information gained about the causal structure. Again, unlike the case of DAGs, where a body of work on experiment design exist, the work in the cyclic setting [23] is few and far in between. This is in part due to the fact that, unlike in DAGs, where an intervention on a subset of the vertices orients all the edges incident to them, in cyclic directed graphs, performing experiments in some cases, neither leads to learning the presence of edges nor orienting them [23].

In this work, we study the problem of causal structure learning from a combination of observational and interventional data in systems governed by linear non-Gaussian structural causal models (SCMs) that may contain *cycles*. Our main contributions are as follows:

- We establish that, under linear non-Gaussian assumptions, the causal graph can be identified from observational data only up to an equivalence class. We further provide a combinatorial characterization of the equivalence class, where each graph corresponds to a perfect matching in a bipartite graph (Section 4.2). Additionally, we show that the condensation graph, or Strongly Connected Components (SCCs), of the true causal graph can also be identified (4.1).

- The bipartite representation allows us to analyze how atomic interventions constrain the space of causal graphs. Specifically, we show that each intervention reveals one edge of the true matching and eliminates all incompatible graphs from the equivalence class (Section 5). Therefore, we can formulate the optimal experiment design problem as an adaptive stochastic optimization over the space of equivalence classes, using a reward function that quantifies the number of eliminated graphs following an intervention. We prove that this reward function is adaptive submodular and hence a greedy policy has provably near-optimal performance guarantees for intervention design (Section 6).

- To address the computational challenge of reward estimation, we propose a sampling-based estimator based on random matchings and provide a theoretical analysis of its bias and concentration properties (Section 7).

- Experiments show that our adaptive strategy outperforms other heuristic methods and closely matches the feedback vertex set (FVS) lower bound (see more details in Section **??**).

## 2   Related Work

Causal discovery is concerned with learning the underlying causal graph which encodes both the existence and direction of edges among variables of interest in a system. From observational data alone, the causal graph can be recovered only up to its Markov equivalence class (MEC). [29] provided necessary and sufficient conditions for Markov equivalence of directed graphs (DGs) and proposed an algorithm for structure learning up to the MEC [28]. [24] extended the applicability of a classic algorithms for constraint-based causal discovery (i.e., FCI [32]) to cyclic DGs and showed that it can recover the structure up to the MEC in this more general setting.

To address latent confounding and nonlinear mechanisms, [6] introduced $\sigma$-connection graphs ($\sigma$-CGs)—a flexible class of mixed graphs—and developed a discovery algorithm that handles both latent variables and interventional data. [10] focused on distributional equivalence for linear Gaussian models, providing necessary and sufficient conditions and proposing a score-based learning method. In the context of linear non-Gaussian models, [21] extended the Independent Component Analysis (ICA)-based approach of [31] to cyclic graphs to learn the equivalence class from the observational data.

The research on experiment design in acyclic models has focused on various aspects, including cost minimization and efficient learning under budget constraints. [5] provided worst-case bounds on the number of experiments to identify the graph. [14] introduced adaptive and exact non-adaptive algorithms for singleton interventions. [13] proposed optimal one-shot and adaptive heuristics to minimize undirected edges. [30] offered lower bounds for experiment design using separating systems, and [20] introduced a stage-wise approach in the presence of latent variables. In cost-aware settings, [19] developed an optimal algorithm for variable-cost interventions without size constraints. [12] and [33] designed approximation algorithms for trees and general graphs, respectively.

Fixed-budget design was initiated by [8] with a greedy Monte Carlo-based approach. [9] improved MEC sampling efficiency, and [7] developed an exact experiment design algorithm for tree structures. [2] proposed an exact method to enumerate DAGs post-intervention. [36, 37] showed that MEC counting and sampling can be done in polynomial time. Bayesian methods include the adaptive submodular algorithm [1] and its extension [34] to optimize both intervention targets and their assigned values.

To the best of our knowledge, only a handful of work on experiment design in cyclic graphs exists. For general SCMs, [23] proposed an experiment design approach that can learn both cyclic and acyclic graphs. They provided a lower bound on the number of experiments required to guarantee the unique identification of the causal graph in the worst case, showing that the proposed approach is order-optimal in terms of the number of experiments up to an additive logarithmic term. Our approach differs from [23] in three key aspects. First, their framework is non-adaptive, that is, all interventions are selected in advance based solely on observational data. In contrast, our method is adaptive as each experiment is selected based on the outcomes of prior interventions. Second, [23] allow intervening on multiple variables in a single experiment. Although they also considered the setup where the number of interventions per experiment is bounded, this bound should be greater than the size of the largest strongly connected component in the graph minus one. By contrast, our framework limits each experiment to a single-variable intervention, making it more practical in settings where fine-grained or limited interventions are preferred. Third, the approach in [23] is designed for general SCMs and relies on conditional independence testing to recover the causal structure. In our work, we focus specifically on linear non-Gaussian models, which enables us to use results from ICA to infer the underlying graph.

## 3  Preliminaries and Problem Formulation

Consider a *Structural Causal Model* (SCM) $\mathfrak{C} := (\mathbf{S}, P_{\mathbf{e}})$, where $\mathbf{S}$ consists of $n$ structural equations and $P_{\mathbf{e}}$ is the joint distribution over exogenous noise terms [26]. In a linear SCM (LSCM), the structural equations take the following form:

$$\mathbf{x} = W\mathbf{x} + \mathbf{e}, \tag{1}$$

where $\mathbf{x} \in \mathcal{X} \subseteq \mathbb{R}^n$ denotes the vector of observable variables, $\mathbf{e} \in \mathcal{E} \subseteq \mathbb{R}^n$ represents the vector of independent exogenous noises, and $W \in \mathbb{R}^{n \times n}$ is the matrix of linear coefficients. The directed graph $G = (V, E)$ induced by the SCM captures the causal structure: node $i$ corresponds to the variable $x_i$, and for any nonzero coefficient $W_{ij} \neq 0$, a directed edge $(j, i) \in E$ is drawn. The binary adjacency matrix $B_G \in \{0, 1\}^{n \times n}$ is defined as $[B_G]_{ij} = \mathbb{1}_{\{W_{ij} \neq 0\}}$, and the "free parameters" are the nonzero entries $\operatorname{supp}(W) = \{(i, j) : W_{ij} \neq 0\}$. We impose the following assumptions:

**Assumption 3.1.** The model has no self-loops: $W_{ii} = 0$ for all $i \in [n]$.
*Remark.* This assumption can be made without loss of generality, as any self-loop can be algebraically removed by reparametrizing the structural equations. For a detailed explanation, see [22].

**Assumption 3.2.** The matrix $I - W$ is invertible.

**Assumption 3.3.** The components of $\mathbf{e}$ are jointly independent, and at most one component is Gaussian. That is,

$$P_{\mathbf{e}} \in \mathcal{P}(\mathcal{E}) := \left\{ P_{\mathbf{e}} : P_{\mathbf{e}} = \prod_{i=1}^n P_{e_i}, \text{ with at most one } P_{e_i} \text{ Gaussian} \right\}.$$

Under these assumptions, the model admits a unique solution: $\mathbf{x} = (I - W)^{-1}\mathbf{e} = A\mathbf{e}$. The observational distribution $P_{\mathbf{x}}$ is thus given by the push-forward measure $P_{\mathbf{x}} = (I - T_W)_{\#}^{-1}(P_{\mathbf{e}})$, where $T_W$ denotes the linear map associated with $W$.

We consider *perfect interventions*, in which the causal mechanism for a variable $x_i$ is replaced with an exogenous noise $\tilde{e}_i$, removing all incoming edges to $x_i$. The interventional SCM is given by:

$$\mathbf{x} = W^{(i)}\mathbf{x} + \mathbf{e}^{(i)}, \tag{2}$$

where the $i$-th row of $W^{(i)}$ is zeroed out and the noise in the $i$-th row of $\mathbf{e}$ is replaced with the new independent noise $\tilde{e}_i$. We denote the new noise vector by $\mathbf{e}^{(i)}$. While $(I - W^{(i)})$ might not always

be invertible, we assume that the probability it becomes singular is zero under mild randomness conditions on the rows of $W$. Thus, we treat $(I - W^{(i)})$ as invertible so that the interventional distribution $P(\mathbf{x}|\mathrm{do}(x_i)) = (I - T_{W^{(i)}})_{\#}^{-1}(P_{\mathbf{e}^{(i)}})$ is well defined.

In interventional structure learning, we perform $K$ distinct experiments to reduce uncertainty over the underlying causal graph. Each experiment involves a *perfect intervention* on a single variable. We denote the set of intervention targets by $\mathcal{I} = \{i_1, i_2, \ldots, i_K\} \subseteq [n]$, where $i_j \in [n]$ indicates that the $j$-th experiment intervenes on variable $x_{i_j}$. In the single-intervention setting we consider, each experiment replaces the structural assignment of the target variable with a new exogenous noise term (cf. Equation 2). As a result, data from the $j$-th experiment is drawn from the interventional distribution $P(\mathbf{x}|\mathrm{do}(x_{i_j}))$.

Each such experiment provides partial information about the true structural matrix $W$ by revealing the causal parents of the intervened variable. The overall objective is to combine observational data with strategically designed interventions to eliminate incorrect graph candidates and recover the true causal structure with minimal experimentation.

Before formulating an optimization strategy for experiment design, we first investigate the extent to which the observational distribution constrains the underlying causal graph and matrix $W$. This characterizes the *observational equivalence class* of graphs, that is, the set of causal structures that cannot be distinguished based on observational data alone.

# 4    Distribution Equivalent Graphs

We aim to characterize the class of linear structural causal models (LSCMs) and their associated graphs that generate the same distribution over observable variables. There are two related but distinct notions in the literature:

- **Distribution-entailment equivalence**, which considers whether two parameterized LSCMs yield the same observational distribution.

- **Distribution equivalence**, which concerns whether two graph structures admit the same set of observable distributions across all compatible LSCMs.

These concepts are formally defined and compared in Appendix A. In our setting—linear SCMs with non-Gaussian noise—the two notions coincide due to identifiability guarantees from ICA. For a detailed discussion, see also Lacerda et al. [22].

*In the remainder of this section, we focus on a graph operation that connects distribution-equivalent graphs.*

**Definition 4.1** (Cycle Reversion). Let $C$ be a cycle in the graph $G$. The **cycle reversion (CR)** operation involves swapping the rows of each member of $C$ in $I + B_G$ with the row corresponding to its subsequent node in the cycle. This operation reverses the direction of the cycle $C$. Moreover, any edge from a node outside $C$ that was originally a parent of some $X \in C$ will now point to the predecessor of $X$ in the original cycle $C$ (see Figure 1a).

Propositions A.3 and A.4 (see Appendix A) shows that distribution-equivalent graphs can be transformed into one another through a sequence of cycle reversions.

## 4.1    Strongly Connected Components

In this section, we examine an important property that remains invariant under cycle reversion. To establish this, we first define strongly connected components (SCCs) in directed graphs.

**Definition 4.2** (Strongly Connected Component (SCC)). Two vertices $u$ and $v$ in a directed graph $G$ are said to be strongly connected if there are directed paths from $u$ to $v$ and from $v$ to $u$. This defines an equivalence relation on the vertex set, whose equivalence classes are called *strongly connected components* (SCCs). Each SCC is a maximal subgraph in which every pair of vertices is strongly connected, and no additional vertex from $G$ can be added without violating this property. The collection of SCCs forms a partition of the vertex set of $G$. See [4, Section 22.5]

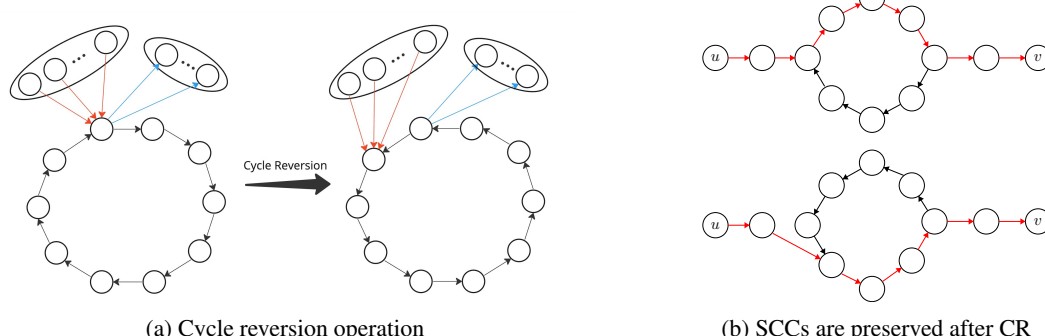

(a) Cycle reversion operation      (b) SCCs are preserved after CR

Figure 1: (a) Graph representation of cycle reversion. (b) The figure highlights representative nodes from an SCC to show that their reachability is maintained, although the full SCC is not depicted.

**Definition 4.3** (Condensation Graph). The *condensation graph* of a directed graph $G$ is obtained by contracting each SCC into a single vertex and mapping all edges between SCCs to a single edge. The resulting graph is a directed acyclic graph (DAG).

**Theorem 4.4.** [1] *Strongly connected components remain unchanged in distribution-equivalent graphs.*

Figure 1b illustrates the core idea behind the proof by showing how cycle reversion operations preserve strongly connected components. The following corollary is a direct consequence of the above theorem and the definition of the condensation graph in 4.3.

**Corollary 4.5.** *The condensation graphs of all distribution-equivalent graphs are identical.*

Additionally, note that for a subset of vertices in the condensation graph that correspond to single vertices in the original graph, their corresponding row in the recovered matrix via *ICA* can be identified. This is because cycles only occur within an SCC, and if an SCC consists of a single vertex, no cycle passes through it, eliminating any *CR* ambiguity. Therefore, the corresponding row can be determined. Furthermore, if the original graph is acyclic, all its SCCs contain only one vertex, meaning that row permutation ambiguity is entirely resolved. As a result, the original data-generating graph can be inferred purely from observational data, a result previously established in Linear Non-Gaussian Acyclic Models (LiNGAM) in [31].

### 4.2 Matching in Bipartite Graphs

Graphs in a distribution-entailment equivalence class correspond to perfect matchings in a bipartite graph. Given $I - W_{\text{ICA}}$, recovered via ICA from observational data, any equivalent graph can be written as $P_\pi D(I - W_{\text{ICA}})$, where $P_\pi$ is a row permutation matrix such that its diagonal entries remain nonzero.

Define a bipartite graph $G_b = (\{r_1, \ldots, r_n\}, [n], E)$, where each node $r_i$ represents the $i$-th row of the matrix $I - W_{\text{ICA}}$, and an edge $(r_i, j) \in E$ exists if and only if $W_{\text{ICA}}[i, j] \neq 0$. Each valid permutation $\pi$ defines a perfect matching $r_i \mapsto \pi(i)$, and vice versa. Thus, distribution-equivalent graphs correspond to row permutations consistent with matchings in $G_b$ (see an example in Appendix A.4).

This bipartite view provides a combinatorial handle on equivalence classes and forms the foundation for analyzing how interventions restrict the matching space.

## 5 Interventional Distribution

We now investigate how interventional data, when combined with observational data, provides additional knowledge about the causal structure. In particular, we show that an intervention on a single variable enables the identification of its causal parents.

---

[1] All proofs are provided in the appendix.

**Proposition 5.1.** *Given the observational distribution generated by the model in* (1) *and the inter-ventional distribution from model* (2)*, the $i$-th row of the matrix $W$ can be recovered. Consequently, the parents of $x_i$ and their causal effects are identifiable.*

## 5.1 Recovering an Edge of the True Matching in the Bipartite Graph

Proposition 5.1 has a natural interpretation in the bipartite graph of 4.2. Recall that each perfect matching in the bipartite graph $G_b = (\{r_1, \ldots, r_n\}, [n], E)$, with edges $(r_i, j) \in E$ iff $W_{\mathrm{ICA}}[i, j] \neq 0$, corresponds to a valid permutation matrix $P_\pi$ defining a distribution-equivalent graph.

Intervening on variable $x_i$, identifies the row corresponding to $x_i$. In the bipartite graph, this identifies the matching edge $(r_{\pi^{-1}(i)}, i)$ corresponding to the true permutation. This eliminates ambiguity for row $r_{\pi^{-1}(i)}$ and restricts the possible options for matching the remaining vertices in the bipartite graph (see an example in Appendix B.2).

# 6 Adaptive Experiment Design

In previous sections, we showed that observational data alone allows us to recover the causal structure only up to an equivalence class of graphs, denoted by $\mathcal{G}_{\mathrm{eq}}$. Each member of this class corresponds to a different row permutation of the matrix $I - W_{\mathrm{ICA}}$, which is itself associated with a valid matching in the bipartite graph introduced in Section 4.2.

To uniquely identify the causal structure, we need to perform interventions. As demonstrated in Proposition 5.1, an intervention on variable $x_i$ reveals the row corresponding to $x_i$, thereby identifies the correct matching edge $(r_{\pi^{-1}(i)}, i)$ in the bipartite graph. Each such intervention removes all graphs from $\mathcal{G}_{\mathrm{eq}}$ that are incompatible with the identified edge, shrinking the equivalence class.

We now define an **adaptive** experiment design framework tailored to this setting. Our objective is to strategically select a sequence of intervention targets to maximally reduce the size of the equivalence class $\mathcal{G}_{\mathrm{eq}}$, with the goal of identifying the true causal graph $G^*$ using as few interventions as possible.

Let $\mathcal{I} = \{i_1, i_2, \ldots, i_K\} \subseteq [n]$ be the set of selected variables for intervention. For each $i_j \in \mathcal{I}$, we collect the interventional data from the distribution $\mathbf{x}^{(i_j)} \sim P(\mathbf{x}|\mathrm{do}(x_{i_j}))$, and perform ICA to estimate the corresponding interventional matrix $I - W^{(i_j)}$. By comparing this estimate with the observational estimate $I - W_{\mathrm{ICA}}$, we recover the row of $x_{i_j}$ and hence the corresponding matching edge. We then eliminate all graphs in $\mathcal{G}_{\mathrm{eq}}$ that are incompatible with this edge.

Let $\Omega := \mathcal{G}_{\mathrm{eq}}$ denote the initial equivalence class. For a given intervention set $\mathcal{I} \subseteq [n]$ and a true graph $\phi \in \Omega$, let $\Omega^{(\mathcal{I}, \phi)} \subseteq \Omega$ denote the set of all graphs that are compatible with the result of interventions on variables in $\mathcal{I}$. The objective is to find the set $\mathcal{I}$ that minimizes the size of the reduced equivalence class $|\Omega^{(\mathcal{I}, \phi)}|$. In the absence of a prior on the true graph $\phi \in \Omega$, we consider a uniform prior over $\Omega$. This leads to an expected-size criterion for selecting the next intervention.

## 6.1 Adaptive Stochastic Optimization

Our aim is to eliminate as many candidate graphs as possible *adaptively*, by choosing each intervention based on the outcomes observed so far. Let the unknown true graph be a random variable drawn uniformly from the set of observationally equivalent graphs, i.e., $\Phi \sim \mathrm{Unif}(\Omega)$.

Following the *Adaptive Stochastic Optimization* framework [11], we define:

- A *policy* $\pi$ is an adaptive strategy that, at each step $t$, selects an intervention $I_t = \pi\big((i_1, O_1), \ldots, (i_{t-1}, O_{t-1})\big)$, based on the history of past interventions $i_1, \ldots, i_{t-1}$ and their observed outcomes $O_1, \ldots, O_{t-1}$. Here, each outcome $O_s$ corresponds to the information gained from intervening on variable $i_s$—for example, the index $\pi^{-1}(i_s)$ of the true row recovered via ICA, or equivalently, the bipartite matching edge $(r_{\pi^{-1}(i_s)}, i_s)$ that was identified.

- For a given realization $\phi \in \Omega$ and policy $\pi$, let $\mathcal{I}(\pi, \phi) = \{i_1, i_2, \ldots, i_K\}$ be the (random) set of variables on which $\pi$ intervenes before terminating (subject to a budget of $K$ interventions).

- Define the *reward function* as

$$f(\mathcal{I}, \phi) := |\Omega| - |\Omega^{(\mathcal{I}, \phi)}|, \tag{3}$$

  where $\mathcal{I} \subseteq [n]$ denotes the set of intervened variables, and $\Omega^{(\mathcal{I}, \phi)} \subseteq \Omega$ is the set of graphs that agree with $\phi$ on the outcomes of all interventions in $\mathcal{I}$.

The *utility* of a policy $\pi$ is the expected reward:

$$F(\pi) = \mathbb{E}_{\Phi \sim \mathrm{Unif}(\Omega)} \left[ f\big(\mathcal{I}(\pi, \Phi), \Phi\big) \right]. \tag{4}$$

Our goal is to find an adaptive policy $\pi$, subject to a budget of $K$ interventions, that maximizes $F(\pi)$.

## 6.2 Adaptive Monotonicity and Submodularity

We now review some relavent definitions from the adaptive submodularity framework [11], and show that our reward function satisfies these properties.

**Definition 6.1** (Universe and Random Realization). Let $\Omega$ be the finite set of all possible true graphs (realizations) consistent with the observational data. We treat the true graph as a random variable $\Phi \sim \mathrm{Unif}(\Omega)$.

**Definition 6.2** (Partial Realization). Let $\mathcal{E} = \{1, 2, \ldots, n\}$ denote the set of variables eligible for intervention, which, in most cases, coincides with the full set of observed variables, and let $O$ denote the set of possible outcomes from each intervention. A *partial realization* $\psi$ is a function

$$\psi : \mathrm{dom}(\psi) \to O, \quad \mathrm{dom}(\psi) \subseteq \mathcal{E},$$

where $\psi(i) \in O$ records the observed outcome (e.g., recovered matching edge) for intervention $i$. We write $\Phi \sim \psi$ to denote the posterior distribution over realizations conditioned on consistency with all observations in $\psi$, i.e.,

$$\Pr[\Phi = \phi | \Phi \sim \psi] \propto \begin{cases} 1 & \text{if } \phi \text{ agrees with } \psi, \\ 0 & \text{otherwise.} \end{cases}$$

**Definition 6.3** (Conditional Expected Marginal Benefit). Let $f : 2^{\mathcal{E}} \times \Omega \to \mathbb{R}_{\geq 0}$ be a reward function and let $\psi$ be a partial realization. For any $v \notin \mathrm{dom}(\psi)$, define the *conditional expected marginal benefit* as

$$\Delta(v|\psi) := \mathbb{E}\left[ f\big(\mathrm{dom}(\psi) \cup \{v\}, \Phi\big) - f\big(\mathrm{dom}(\psi), \Phi\big) \mid \Phi \sim \psi \right]. \tag{5}$$

**Definition 6.4** (Adaptive Monotonicity). A reward function $f$ is *adaptive monotone* if

$$\Delta(v|\psi) \geq 0 \tag{6}$$

for every partial realization $\psi$ and item $v \notin \mathrm{dom}(\psi)$.

**Definition 6.5** (Adaptive Submodularity). A reward function $f$ is *adaptive submodular* if, for all partial realizations $\psi \subseteq \psi'$ and all $v \notin \mathrm{dom}(\psi')$,

$$\Delta(v|\psi) \geq \Delta(v|\psi'). \tag{7}$$

**Theorem 6.6.** *Let $f(\mathcal{I}, \phi)$ be the number of graphs eliminated after performing interventions $\mathcal{I}$ under realization $\phi$, as defined in Equation (3). Then $f$ is adaptive monotone and adaptive submodular with respect to the uniform prior distribution over realizations.*

*Let $F(\pi) = \mathbb{E}_{\Phi \sim \mathrm{Unif}(\Omega)} \left[ f\big(\mathcal{I}(\pi, \Phi), \Phi\big) \right]$ denote the expected number of eliminated graphs under an adaptive policy $\pi$, as defined in Equation (4). Then the adaptive greedy policy achieves a $(1 - 1/e)$-approximation to the optimal value of $F$.*

## 7 Estimating the Reward Function by Sampling

Based on Theorem 6.6, at each step (after observing the results of previous interventions), we greedily intervene on the variable that maximally reduces the size of the remaining equivalence class. Thus, it is sufficient to compute the conditional expected marginal benefit $\Delta(v|\psi)$ for each candidate variable $v \notin \mathrm{dom}(\psi)$.

Let $\phi$ be any realization consistent with $\psi$, and define $N = |\Omega^{(\mathrm{dom}(\psi),\phi)}|$. In the bipartite-matching representation of the problem, the column vertex $v$ in the bipartite graph may be matched to different rows across the candidate graphs in $\Omega^{(\mathrm{dom}(\psi),\phi)}$. Concretely, recall that each graph in this equivalence class corresponds to a perfect matching in the bipartite graph $G_b$, whose left nodes are rows $r_1, \ldots, r_n$, and whose right nodes are column indices $1, \ldots, n$. Before intervening on variable $v$, the right-hand node $v$ may be matched to different row nodes $r_{z_1}, \ldots, r_{z_k}$ across the $N$ graphs. Denote these distinct row indices by $z_1, z_2, \ldots, z_k$. Therefore, the set of candidate edges for node $v$ is

$$\{ (r_{z_i}, v) \; : \; i = 1, \ldots, k \}.$$

Each graph $g \in \Omega^{(\mathrm{dom}(\psi),\phi)}$ selects exactly one of these edges in its matching. Let

$$n_i = \left| \left\{ g \in \Omega^{(\mathrm{dom}(\psi),\phi)} \; : \; g \text{ matches } v \text{ to } r_{z_i} \right\} \right|, \qquad \sum_{i=1}^{k} n_i = N.$$

Therefore, prior to an intervention, the probability that $v$ is matched to row $r_{z_i}$ is $p_i := n_i/N$. Once we intervene on $v$ and recover its true row, we identify the correct edge $(r_{z_i}, v)$ and eliminate all graphs that do not contain this edge. If edge $(r_{z_i}, v)$ is observed, exactly $n_i$ graphs survive, and the number of eliminated graphs is $N - n_i$. Hence, the conditional expected marginal benefit under $\psi$ is:

$$\Delta(v|\psi) = \sum_{i=1}^{k} p_i(N - n_i) = N \sum_{i=1}^{k} p_i(1 - p_i). \tag{8}$$

To compute $\Delta(v|\psi)$ via Equation (8), we need to know the full equivalence class $\Omega^{(\mathrm{dom}(\psi),\phi)}$ (This is necessary for evaluating $n_i$ and $N$). Since explicitly enumerating all graphs in this class is computationally infeasible due to the exponential number of perfect matchings, we estimate $\Delta(v|\psi)$ by sampling from $\Omega^{(\mathrm{dom}(\psi),\phi)}$.

Let $\mathbf{p} = [p_1, \ldots, p_k]^T$ and define the normalized marginal benefit as $L(\mathbf{p}) := \sum_{i=1}^{k} p_i(1 - p_i)$. Although $\mathbf{p}$ is not directly accessible, we assume a **sampling oracle** $\mathcal{S}$ that returns i.i.d. samples from $\mathbf{p}$. This corresponds to sampling a perfect matching uniformly at random and observing the edge incident to $v$. A single batch of such samples allows us to estimate $L(\mathbf{p})$ for all intervention candidates.

The existence of a polynomial-time approximate uniform sampler is ensured by the FPRAS of Jerrum, Sinclair, and Vigoda [16]. While their method offers theoretical guarantees, we also consider faster heuristic alternatives in practice. For the sake of analysis, it suffices to assume access to such a sampler.

Given samples $X_1, \ldots, X_M \sim \mathbf{p}$, where each $X_j \in [k]$ indicates the row matched to $v$, we construct an empirical estimate $\hat{\mathbf{p}}$ and use it to approximate $\Delta(v|\psi)$.

## 7.1 Statistical Accuracy of Sampling-Based Estimation

We now quantify the accuracy of estimating the marginal benefit $\Delta(v|\psi) = N L(\mathbf{p}) = N \sum_{i=1}^{k} p_i(1 - p_i)$ via empirical sampling.

Let $\mathbf{p} = (p_1, \ldots, p_k)$ be the unknown categorical distribution over candidate rows (i.e., the edges connected to $v$ in the bipartite graph). Let $X_1, \ldots, X_M \overset{\mathrm{iid}}{\sim} \mathbf{p}$, and define the empirical distribution:

$$\hat{\mathbf{p}} = (\hat{p}_1, \ldots, \hat{p}_k), \quad \hat{p}_i := \frac{1}{M} \sum_{j=1}^{M} \mathbf{1}\{X_j = i\}. \tag{9}$$

We use the estimator $L(\hat{\mathbf{p}}) = \sum_{i=1}^{k} \hat{p}_i(1 - \hat{p}_i)$. The following theorem provides an estimation error bound on $|L(\mathbf{p}) - L(\hat{\mathbf{p}})|$.

**Theorem 7.1.** *Given $M$ i.i.d samples $X_1, \ldots, X_M \overset{\mathrm{iid}}{\sim} \mathbf{p}$, for the estimsator $L(\hat{\mathbf{p}})$, we have:*

$$\mathbb{P}\left[ |L(\hat{\mathbf{p}}) - L(\mathbf{p})| \geq \frac{1}{M} + \sqrt{\frac{2}{M} \log\left(\frac{2}{\epsilon}\right)} \right] \leq \epsilon. \tag{10}$$

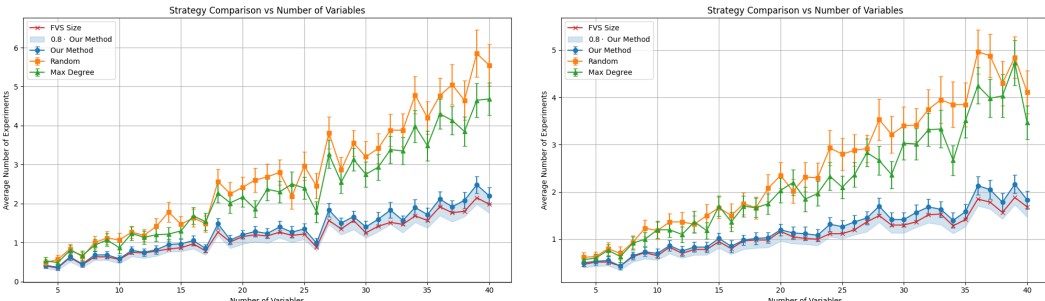

(a) Sample mode: approximate matching sampler.  (b) Exact mode: full enumeration of matchings.

Figure 2: Comparison of intervention strategies assuming an ideal ICA oracle. Our method (Adaptive) consistently performs close to the feedback vertex set (FVS) lower bound.

In particular, picking $\epsilon = \frac{1}{M}$ implies that with probability at least $1 - \frac{1}{M}$, the error satisfies

$$|L(\hat{\mathbf{p}}) - L(\mathbf{p})| \leq \frac{1}{M} + \sqrt{\frac{2 \log(2M)}{M}} = \mathcal{O}\left(\sqrt{\frac{\log M}{M}}\right) = \widetilde{\mathcal{O}}\left(\sqrt{\frac{1}{M}}\right).$$

## 8 Experimental Results

We evaluate the performance of our adaptive experiment design method for identifying causal graphs under the linear non-Gaussian structural causal model assumption. Experiments are conducted on synthetic data generated from Erdős–Rényi random directed graphs, where each possible directed edge (excluding self-loops) is included independently with a fixed probability. To ensure identifiability, we enforce that the resulting matrix $I - W$ is invertible. We measure the average number of interventions required for full causal identification.

### 8.1 Performance with an Ideal ICA Oracle

To evaluate the intervention strategy on its own, isolating it from statistical estimation errors, we assume access to an ideal ICA procedure that returns a row-permuted and scaled version of the true matrix $I - W$. The adaptive method uses a bipartite representation of the graph and samples perfect matchings using two modes: *exact*, where all matchings are enumerated, and *sample*, where a fast greedy heuristic is used (E.2). We compare our adaptive strategy against two baselines: *Random*, which selects a target uniformly at random, and *Max Degree*, which chooses the node with the highest degree in the bipartite graph.

As a theoretical baseline, we compute the *feedback vertex set (FVS)* of each true graph. The FVS size represents a fundamental lower bound on the number of interventions required to break all cycles. Although computing the FVS is NP-hard, we solve it exactly to obtain the best possible benchmark.

Figures 2a and 2b show that our adaptive method consistently outperforms the baselines and operates remarkably close to the intractable FVS lower bound, despite having no knowledge of the true graph structure. This demonstrates the near-optimality of our greedy approach in an ideal setting. For more details, please refer to E.

### 8.2 Robustness in a Practical Setting with Finite-Sample ICA

To assess the practical viability of our method, we evaluate its performance in a more realistic setting without an ideal ICA oracle. Instead, we apply the `FastICA` algorithm [15] to finite samples generated from both observational and interventional distributions. This introduces estimation noise, which can corrupt the recovered matrices. To handle these challenges, we introduce two key algorithmic modifications for robustness: *adaptive thresholding* of matrix entries and a *safe matching procedure* to prevent incorrect row assignments. A detailed description of these modifications is provided in Appendix E.3.

In this setting, our adaptive strategy continues to exhibit a performance trend similar to the ideal case, consistently outperforming the Random and Max Degree baselines and remaining remarkably close to the FVS lower bound (see Figure 6 in Appendix E.3), confirming its effectiveness even under estimation noise of ICA.

Furthermore, to analyze the quality of the recovered structure, we measured the relative error ($\varepsilon_{\mathrm{rel}} = \frac{\|\widehat{W} - W^*\|_F}{\|W^*\|_F}$) between the estimated matrix $\widehat{W}$ and the ground-truth $W^*$. Our result, presented in Appendix E.3, shows that the algorithm achieves a high-fidelity recovery of the true causal structure in the vast majority of runs.

## 9   Extensions

Our core framework focuses on adaptive, single-variable, perfect interventions in linear non-Gaussian models. Here, we briefly discuss how the proposed method can be extended to more general settings. We defer detailed derivations and proofs to Appendix F.

### 9.1   Multi-Node Interventions

The proposed method can be generalized to handle simultaneous interventions on a set of variables $E = \{i_1, \ldots, i_t\}$. Such an intervention is always informative, as it localizes the row-permutation ambiguity to the intervened set $E$ and effectively disambiguates it from the rest of the graph. Furthermore, if the subgraph induced by the variables in $E$ is acyclic, the internal cycle reversion ambiguity within $E$ is fully resolved. This allows for the unique recovery of the true row for each variable in $E$, enabling the safe parallelization of experiments to accelerate the causal discovery process. We provide a detailed analysis in Appendix F.1.

### 9.2   Imperfect Interventions

Our approach is also robust to imperfect interventions. The core requirement for identifying a variable's corresponding row is that the intervention sufficiently perturbs its causal mechanism. As long as the post-intervention row is distinguishable from the set of observational rows after ICA estimation, the intervention is informative. This condition holds for some realistic noisy or incomplete intervention types, enhancing the practical applicability of our method. Further details are discussed in Appendix F.2.

### 9.3   Generalization to Non-linear Models

While we focused on the linear case, our experimental design strategy could be extended to non-linear structural causal models. Recent advances in non-linear ICA can recover the Jacobian of the model's inverse function, whose sparsity pattern reveals the underlying causal graph structure up to the same permutation ambiguity found in the linear setting [27]. Our bipartite matching formulation and reward function could then be applied to resolve this ambiguity. We outline this potential generalization in Appendix F.3.

## 10   Conclusion

We introduced a framework for causal structure learning in linear non-Gaussian models with cycles, leveraging a combinatorial characterization of equivalence classes via bipartite matchings. By formalizing experiment design as an adaptive submodular optimization problem, we developed a near-optimal greedy policy that incrementally resolves causal ambiguity through targeted interventions. Our sampling-based estimator enables practical implementation without exhaustive enumeration. Empirical results confirm that our method recovers the true graph with few interventions, often matching the feedback vertex set lower bound, despite having no access to the true structure.

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
