# OpenReview forum: "Near-Optimal Experiment Design in Linear non-Gaussian Cyclic Models"
_NeurIPS.cc/2025/Conference — NeurIPS 2025 spotlight_

### Official Review · Reviewer_p51C · 2025-07-01

**Clarity:** 3
**Significance:** 3
**Originality:** 2
**Rating:** 5
**Confidence:** 3

**Summary:**

The authors characterize graphically the equivalence class of graphs learnable from observational data as a bipartite graph where each node represents the ith row of the matrix (I-W) and an edge in the matching only exists if the jth entry of W is nonzero. Each valid permutation over rows defines a perfect matching. So distribution-equivalent graphs are row permutations consistent with matchings in the bipartite graph.

**Questions:**

Questions:

In appendix E the algorithm seems to do less favorably on dense graphs with a max of 60 nodes. Why is that? It seems the graphical structure might have something to do with it since all the methods appear to be doing nearly the same (qualitatively the algorithm is performing like max degree). Since erdos renyi graphs are so well-studied, is there some graphical statistic that could be used to explain when the adaptive algorithm will in particular do better than a max degree heuristic?

**Ethical Concerns:**

["NO or VERY MINOR ethics concerns only"]

**Final Justification:**

I have read the reviews and rebuttal and maintain my score of accept. I am satisfied with the rebuttal response.

**Quality:**

3

**Strengths And Weaknesses:**

strengths:

- significance: the authors nicely set up the technical difficulty of learning with observational and interventional data with cycles.
- quality: the paper contributes nice constructions and characterizations to this problem. The graphical characterization is nice, and the characterization of what is learned by intervening is nice. (I am not enough of an expert to know whether these characterizations are immediate or not). This reduces the combinatorial search problem quite a bit. The authors construct an adaptive search procedure with appealing properties, given access to a sampler over the space of perfect matchings, which is a relatively friendly object. The paper is a nice end-to-end procedure (given the output of ICA) that relates to fairly classical graphical ideas.

weaknesses:

- relevance: it's not so clear when causal discovery in cyclic graphs is relevant, what are the new potential real-world cases of causal discovery where you have both observational data and the capacity to run interventions that this can enable? the simulations are on random graphs and therefore not illuminating of the potential real-world relevance.


- significance:

-- The paper is a good characterization of the "interventional" part of causal discovery under observational+interventional data with cycles. It's not how the characterizations and approach of this paper are connected to the "observational" part of learning, ie from finite samples. The paper could be improved by exploring this front.
-- The development of the adaptive algorithm enjoys a lot of relatively tractalbe structure. The objective is to find the set I that minimizes the size of the reduced equivalence class |Ω(I,ϕ)|. However, a lot of later characterizations seem to rely quite heavily on the uniform prior assumption. For example, I'm a little confused by this, "Therefore, prior to an intervention, the probability that v is matched to row rzi is pi = ni/N.", is this at the first step with the uniform prior, or at intermediate later timesteps as well?

The empirical analysis is a good proof of concept but not super thorough

- clarity

Overall good, but the paper could benefit from some explanations at certain places. I will put some questions here because they are more for expository purposes (and not really questions that I need answers to "in order to improve my score" as in the other box.)

Questions:
- what is the SCC that is preserve in Fig 4b?
- Why does intervening on variable xi, 218 identifies the row corresponding to xi. how should i interpret this / why is this different than interventions in the acyclic case? ]
- What is the interpretation of (I-W) and its rows/permutations?
- this is confusing:
posterior distribution over realizations conditioned on consistency with all
274 observations in ψ
- "Figures 2a and 2b show the average number of interventions under each strategy."
To do what? Recover the full graph? Exhaust the nodes? What feedback does the algorithm get to know that it's discovered the whole graph?

---

> ### Author Rebuttal · Authors · 2025-07-31
>
> * Topic 1: Real-World Relevance of Cyclic Causal Discovery: In many real-world systems, feedback loops among variables play a crucial role in maintaining stability. These loops often induce cycles in the system's causal graph, particularly when temporal dynamics are sampled at a low rate or when modeling the system at equilibrium (Bongers et al., 2021). A prominent example is gene regulatory networks (GRNs), where sets of biological regulators interact to control the expression levels of proteins. A GRN can be represented by a directed graph, where nodes correspond to genes, and a directed edge from gene X to gene Y indicates that activation of gene X may directly promote or inhibit gene Y. An intervention on a gene typically involves "knocking out" that gene, which corresponds to removing all incoming edges to it. The objective is to recover the underlying causal structure of the GRN using both observational data and strategically designed interventions.
> * Topic 2: Uniform Prior Assumption: The uniformity assumption holds not only at the first step, but also throughout all intermediate steps of the adaptive process. At each round, the algorithm maintains the current interventional-equivalence class $\Omega_t$, which consists of the set of graphs consistent with all observations and interventions so far.
> We assume a uniform prior over the initial class $\Omega_0$, and the posterior over the current class $\Omega_t$ remains uniform as it is simply a filtered subset of $\Omega_0$ consistent with the history. Note that we do not impose a new uniform prior at each step; rather, the uniform distribution over $\Omega_t$ is inherited from the initial uniform prior and updated by eliminating graphs inconsistent with observed interventional outcomes. This induces a categorical distribution over matchings of any variable $v$, based on how often each row $r_i$ appears as a match to $v$ across the remaining graphs. Specifically, if $\Omega_t = \lbrace g_1,\dots,g_{N_t} \rbrace$, and $n_i$ of those graphs match $v$ to row $r_{z_i}$, then the probability that $v$ is matched to $r_{z_i}$ is $p_i = \frac{n_i}{N_t}.$
> This logic is valid at the first step (where $\Omega_0$ is the full equivalence class) and at every later step, since the posterior is uniform over the consistent subset of graphs $\Omega_t$. The structure of our bipartite matching formulation allows this uniformity to be efficiently updated after each intervention, preserving tractability throughout the algorithm.
> * Topic 3: Clarification of Technical Details
>     * What SCC is being preserved in Fig. 1b?
>     In Fig. 1b, we do not depict the full SCC; rather, we illustrate a subset of nodes from a single SCC to emphasize a key property: cycle reversion preserves the SCC structure. That is, even after applying cycle reversion, any two nodes that belonged to the same SCC remain mutually reachable, and therefore still lie in the same SCC.
> We agree that the caption “SCCs are preserved after CR” and the limited scope of the illustration might be slightly misleading, as the figure shows only a portion of the SCC. A more accurate caption would clarify that the figure highlights two representative nodes from an SCC to demonstrate that their mutual reachability remains intact after reversion. We will revise the caption accordingly in the final version to avoid this ambiguity.
>     * Why does intervening on variable $x_i$ identify its row?
>     The key idea is that, in linear **cyclic models** with non-Gaussian noise, the observational distribution alone is not sufficient to uniquely identify the causal graph. Instead, it yields the matrix $W$ only up to an unknown row permutation, due to a cycle reversion (CR) ambiguity.
> As stated in Proposition 5.1, intervening on a single variable $x_i$ allows us to recover the true $i$-th row of the causal matrix $W$. The intuition is that the intervention modifies the structural equation of $x_i$ (by cutting its incoming edges), so the corresponding row in $W$ is zeroed out under the intervention. Comparing the ICA results from observational and interventional data reveals exactly which row in the observational matrix has been "removed," thus revealing the true position of row $i$ in $W$.
> In the bipartite matching view, this means that the matching edge $(r_{\pi^{-1}(i)}, i)$ becomes identifiable, and the row $r_{\pi^{-1}(i)}$ is assigned definitively to column $i$ in the true permutation. This effectively fixes one row of the matrix and reduces the uncertainty for subsequent steps of the algorithm.
>       * Why this differs from the acyclic case: In linear **acyclic** models with non-Gaussian noise (i.e., LiNGAM), the causal matrix $W$ is uniquely identifiable from observational data alone due to its triangular structure. No permutation ambiguity arises, and thus interventions are not required to disambiguate the matching. In contrast, cyclic models allow multiple observationally equivalent graphs (related via cycle reversion), and resolving this ambiguity necessitates interventional data.
>     * Interpretation of $I-W$ and its row permutations:
>     The matrix $I - W$ corresponds to the reduced-form representation of a linear structural causal model (SCM). In a general SCM, each variable $x_i$ is defined as a function of its parents: $x_i = f_i(\mathrm{PA}_{x_i}, e_i)$, and under linearity, the system can be written compactly as $x = Wx + e$. Solving for $x$, we obtain the reduced-form expression $x = (I - W)^{-1} e$, which expresses each variable as a linear combination of the exogenous noise terms. This reduced form is particularly useful because it captures the overall dependencies between variables and exogenous noise terms and allows for direct estimation from data using ICA.
> Due to ICA's identifiability only up to row permutations and scaling, the rows of the estimated $I - W$ may appear permuted. These row permutations correspond to different assignments of structural equations to variables. In particular, these permutations arise from **cycle reversion** ambiguities, where feedback loops in the causal graph can be flipped without changing the observational distribution. Our method resolves these ambiguities through targeted interventions.
>     * Confusion about posterior distributions.
>     We agree that the phrasing may be unclear without context. As formalized in our definitions, $\Phi \sim \psi$ denotes the posterior distribution over the set of realizations (causal graphs) that are consistent with all observational and interventional outcomes recorded in the current partial realization $\psi$. Each intervention in $\psi$ provides constraints that eliminate inconsistent graphs from the original equivalence class $\Omega$. The remaining feasible graphs—those that agree with all observations in $\psi$—define a subset $\Omega(\psi) \subseteq \Omega$.
> We assume a uniform prior over $\Omega$, and thus maintain a **uniform posterior** over the remaining consistent graphs in $\Omega(\psi)$. All graphs that contradict any observation in $\psi$ are assigned zero probability. In other words, the expression “posterior distribution conditioned on consistency with $\psi$” simply refers to a uniform distribution over the surviving graphs after filtering out those incompatible with the outcomes observed so far. We will clarify this wording in the final version to improve readability.
>     * Stopping Criterion and Interpretation of Intervention Count in Figures 2a and 2b:
>     Figures 2a and 2b report the average number of interventions required to achieve full causal identification of the underlying graph—i.e., to uniquely recover the true matrix $W$.
> The algorithm receives feedback in the form of structural constraints imposed by each intervention (i.e., fixing one row of $W$). As described in the first line of Algorithm 1, the stopping condition is triggered when the bipartite graph—whose edges represent remaining candidate matchings—is reduced to a **unique perfect matching**. This indicates that the row permutation ambiguity has been fully resolved and that all causal edges have been identified. Thus, the number reported in the figures reflects how many interventions are needed until this condition is met.
>     * Justification of performance on dense graphs:
>     In dense Erdős–Rényi graphs, most variables might have many parents, which leads to a large number of candidate matchings in the bipartite graph. As a result, the posterior distribution $\mathbf{p}_v$ over possible rows for each node $v$ becomes nearly uniform, making the information gain $L(\mathbf{p}_v)$ roughly the same for all nodes. This flattens the adaptive gain landscape and causes our method to behave similarly to simple heuristics like MAX-DEGREE, but still outperform those heuristics.
> In contrast, for sparse graphs, the posteriors $\mathbf{p}_v$ are more peaked (some values are close to 1), so the adaptive strategy can better prioritize "high-information" nodes and terminate faster.
>
> Bongers et al., Foundations of Structural Causal Models with Cycles and Latent Variables, The Annals of Statistics, 2021

---

> > ### Comment · Reviewer_p51C · 2025-08-04
> >
> > Thanks for the rebuttal.
> >
> > A challenge that remains is verifying the uniformity of the posterior distribution. I'm a little confused still by the argument that the posterior is uniform, conditional on the eliminated graphs. Is there a standard result from Bayesian inference on posterios over graphs that could be cited here? Otherwise it could be that the structure of the bipartite matching makes it so that maintaining a uniform distribution, conditional on eliminated graphs, is computationally feasible and easy to do, so there just is a uniform distribution maintained over the remaining graphs.
> >
> > It's not immediately clear why "the uniform distribution over $\Omega_t$ is inherited from the initial uniform prior and updated by eliminating graphs inconsistent with observed interventional outcomes," I'm not seeing obviously why conditioning on the eliminated graphs doesn't change the distribution. After all, don't the eliminated graphs share some kind of structure?
> >
> >  I think either a proof is required, or cite standard results in updating Bayesian posteriors on graphs if this is really intended to be Bayesian updated. Else keep the algorithm the same but don't use the prior/posterior language that implies proper Bayesian updating?

---

> > > ### Author Response · Authors · 2025-08-05
> > >
> > > We thank the reviewer for the valuable comments.
> > > To clarify, our setup involves a finite hypothesis space: the set $\Omega_0$ of all causal graphs consistent with ICA output from the observational distribution. We assume a uniform prior over $\Omega_0$, meaning each graph is equally likely before any intervention.
> > >
> > > Let us denote by $O_0$ the outcome of the observational phase (i.e., the ICA output using observational distribution), and by $O_1, O_2, \dots, O_t$, the outcomes of the first $t$ interventional steps. At round $t$, the current hypothesis class is the set $\Omega_t$ of graphs consistent with all outcomes $(O_0, O_1, \dots, O_t)$.
> > > At each round, the result of an intervention eliminates all graphs inconsistent with the observed outcome. The remaining set $\Omega_{t+1} \subset \Omega_t$ consists of graphs consistent with the new outcome $O_{t+1}$ as well.
> > >
> > > In particular, in our setup, the interventional distribution obtained by intervening on variable $x_i$ is used to recover the $i$th row of the causal matrix via ICA. Any graph that places this recovered row in the correct position—i.e., has a consistent bipartite matching—can produce the same interventional distribution. **Thus, no additional information is revealed about the consistent graphs beyond their inclusion in $\Omega_{t+1}$**. As a result, the posterior remains uniform over $\Omega_{t+1}$.
> > >
> > > This intuition can be made rigorous using Bayes’ rule and induction. Let $\Phi$ be the true (unknown) graph. Suppose that after round $t$, the posterior is uniform over $\Omega_t$, i.e.,
> > > $$
> > > \mathbb{P}[\Phi = G \mid O_0, O_1, \dots, O_t] = \frac{1}{|\Omega_t|}, \quad \text{for all } G \in \Omega_t.
> > > $$
> > > Now, we perform the $(t+1)$th intervention and observe outcome $O_{t+1}$, which filters $\Omega_t$ to $\Omega_{t+1} \subset \Omega_t$. For any $G \in \Omega_t$, Bayes' theorem gives:
> > > $$
> > > \mathbb{P}[\Phi = G \mid O_0, \dots, O_{t+1}] =
> > > \frac{\mathbb{P}[O_{t+1} \mid \Phi = G, O_0, \dots, O_t] \cdot \mathbb{P}[\Phi = G \mid O_0, \dots, O_t]}{\mathbb{P}[O_{t+1} \mid O_0, \dots, O_t]}.
> > > $$
> > > Since the outcome $O_{t+1}$ is deterministic given the true graph, we have:
> > > $$
> > > \mathbb{P}[O_{t+1} \mid \Phi = G, O_0, \dots, O_t] = 1_{\lbrace G \in \Omega_{t+1} \rbrace}
> > > $$
> > > Therefore, for $G \in \Omega_{t+1}$, we obtain:
> > > $$
> > > \mathbb{P}[\Phi = G \mid O_0, \dots, O_{t+1}] =
> > > \frac{1 \cdot \frac{1}{|\Omega_t|}}{\mathbb{P}[O_{t+1} \mid O_0, \dots, O_t]}.
> > > $$
> > > Using the law of total probability:
> > > $$
> > > \mathbb{P}[O_{t+1} \mid O_0, \dots, O_t] = \sum_{G' \in \Omega_t} \mathbb{P}[O_{t+1} \mid \Phi = G', O_0, \dots, O_t] \cdot \mathbb{P}[\Phi = G' \mid O_0, \dots, O_t] = \frac{|\Omega_{t+1}|}{|\Omega_t|}.
> > > $$
> > > Plugging this back in, we conclude:
> > > $$
> > > \mathbb{P}[\Phi = G \mid O_0, \dots, O_{t+1}] = \frac{1}{|\Omega_{t+1}|}, \quad \text{for all } G \in \Omega_{t+1},
> > > $$
> > > and zero otherwise. This completes the inductive step, showing that the posterior remains uniform after each round.
> > >
> > > We emphasize that this use of prior/posterior language aligns with standard finite-space Bayesian updating and matches the terminology adopted in our main reference on adaptive submodular optimization: Golovin and Krause, ``Adaptive Submodularity'' (JAIR 2011).

---

### Official Review · Reviewer_kRFZ · 2025-07-01

**Clarity:** 3
**Significance:** 2
**Originality:** 2
**Rating:** 4
**Confidence:** 2

**Summary:**

This paper tackles causal discovery in linear non-Gaussian cyclic models—systems where  cycles make structure learning tricky. Previous work shows that observational data alone can't fully identify the causal graph; instead, we're left with a set of permutation-equivalent candidates. Authors show that each atomic intervention cuts through the uncertainty by revealing one true edge, pruning incompatible graphs from the equivalence class. The authors frame experiment design as an adaptive stochastic optimization problem, proving their reward function is adaptive submodular and that a greedy policy achieves near-optimal efficiency. Simulations confirm that their method, aided by a sampling estimator, recovers the true causal structure with minimal interventions.

**Questions:**

Questions/Comments for the Authors:

Comparison with Mokhtarian et al. (2023):
Could you compare your proposed approach against Mokhtarian et al. (2023) for the linear case? Demonstrating superior performance for this specific class of models would help readers better appreciate the work’s value.

Assumption 3 (Non-Gaussian Noise):
Why is the restriction to at most one Gaussian noise component (Line 125) necessary? Does the model fail to admit a unique solution if multiple noise components are Gaussian? Could you justify this theoretically or with an example or a reference?

Single-Node Intervention Limitation:
What prevents extending your method to multi-node interventions when such data is available? Could a similar approach be adapted for this case?

Sampling-Based Estimator Concerns:
While you characterize the statistical accuracy of the sampling-based estimator, how does its use affect the optimality of solutions obtained via the greedy policy? Specifically, could estimation errors in the reward function lead to suboptimal intervention sequences?

Experimental Details:
How many samples (M) were used for the sampling-based estimation in your experiments? This parameter’s choice seems critical for balancing accuracy and computational cost.

**Ethical Concerns:**

["NO or VERY MINOR ethics concerns only"]

**Final Justification:**

I thank the reviewer for the detailed response to my questions. Including some discussion from the response will improve the manuscript. I will keep score for the paper.

**Limitations:**

yes

**Paper Formatting Concerns:**

N.A.

**Quality:**

2

**Strengths And Weaknesses:**

This work establishes that linear non-Gaussian cyclic models are identifiable only up to an equivalence class, characterized combinatorially via perfect matchings in bipartite graphs. Furthermore, the authors prove that each intervention reveals one true edge, enabling adaptive experiment design with submodular guarantees, and develop an efficient sampling estimator to overcome the computational challenge of graph enumeration. They also provide experimental validation of the proposed approach, demonstrating near-optimal recovery of causal structure with minimal interventions.

Weaknesses: The problem setup is restricted to linear models and assumes non-Gaussian noise (with at most one Gaussian component). Additionally, the noise components must be independent, which limits practical applicability. The paper also lacks comparison with Ehsan Mokhtarian et al. (2023), which, while focusing on general SCMs, remains a valid baseline for the linear case. Furthermore, the framework limits each experiment to single-variable interventions, potentially restricting applications where multi-node intervention data is available.

---

> ### Author Rebuttal · Authors · 2025-07-31
>
> * Topic 1: Linearity, Non-Gaussianity, and Independent Noises Assumption -
> Linear models are prevalent in many applications, mainly because they can be learned with a moderate sample size and offer simple qualitative interpretations. On the theory side, understanding causal phenomena in a linear model is often a necessary step toward a complete understanding of the given phenomenon, see e.g., (Pearl, 2013). Moreover, in many physical and engineering systems, linear models provide accurate approximations around equilibrium points or typical operating regimes. This local linearity is a well-established principle in control theory and system identification (Åström and Murray, 2008; Ljung, 1999), and supports the use of linear assumptions as a starting point even when the underlying system may be globally nonlinear.
> The non-Gaussianity assumption is a common assumption required in a wide range of literature on causal effect estimation and causal discovery (Shimizu 2014, Hoyer et al. 2008, Salehkaleybar et al. 2020, Kivva et al. 2023, Peters et al. 2017). More specifically, for Gaussian distributions, some causal relationships are indistinguishable from observational data. This issue is well-illustrated by the non-identifiability results for linear Gaussian models from observational data (Peters et al. 2017).
> About independent exogenous noises, this is the fundamental assumption in SCM, and it is often called "principle of independent mechanism". Please refer to Chapter 2 of the textbook (Peterts et al., 2017) for the justification of this assumption.
>
> * Topic 2: Why is the restriction to at most one Gaussian noise necessary?
> This condition is required in linear ICA to identify the mixing matrix up to scaling and permutation (Eriksson & Koivunen, 2004). In our setting, it enables recovery of the rows of $ I-W$. For DAGs, it's also necessary to recover the full causal graph from observational data using ICA-LiNGAM (Shimizu et al., 2006).
>
>
> * Topic 3: Comparison with Prior Work (Mokhtarian et al. 2023) -
> In the submitted paper (lines 93–108), we provided a detailed comparison with Mokhtarian et al. (2023). To summarize the key differences: their framework is non-adaptive—all interventions are selected in advance using only observational data. In contrast, our method is adaptive, choosing each intervention based on the outcomes of previous ones. Additionally, their approach involves multi-variable interventions. While they also consider a bounded setting, the required bound must exceed the size of the largest SCC in the graph minus one. Our method, by default, uses single-variable interventions—though it can be extended to the multi-variable case, discussed in the following—which is often more practical in scenarios with limited or fine-grained intervention capabilities. Finally, their method targets general SCMs and uses CI testing to recover the structure, while we focus on linear non-Gaussian models, allowing us to leverage ICA for causal discovery. A direct empirical comparison is not fair to them, given that our adaptive approach typically needs far fewer interventions than a non-adaptive one.
>
> * Topic 4: Multi-Node Interventions -
> Indeed, our method **can be extended to multi-node interventions** when such data is available. Suppose an experiment is conducted where interventions are simultaneously applied to a set of variables $\mathcal{I}= \lbrace i_1, i_2, \dots, i_t \rbrace$. In this case, applying ICA to both observational and interventional datasets yields matrices $P D (I - W)$ and $P' D' (I - W^{(i_1, i_2, \dots, i_t)})$, where $P, P'$ are permutation matrices and $D, D'$ are diagonal scaling matrices. The matrix $W^{(i_1, \dots, i_t)}$ is the post-intervention weight matrix, obtained by zeroing out the $i_1, \dots, i_t$-th rows of $W$, corresponding to interventions on $\lbrace i_1, i_2, \dots, i_t \rbrace$.
> The difference between $P D (I - W)$ and $P' D' (I - W^{(i_1, \dots, i_t)})$ lies precisely in the $t$ rows corresponding to the interventions.
> By comparing the two matrices, we can identify the $t$ unmatched rows and each row belongs to one of the indices in $\lbrace i_1, i_2, \dots, i_t \rbrace$. This turns the global row-permutation problem into a **local one** over the intervened set. More importantly, such an intervention effectively breaks the cycle reversion ambiguity between the intervened set $\mathcal{I}$ and the rest of the graph, isolating the two. As a result, the permutation ambiguity is constrained separately within each set.
> In the multi-node intervention case, it is needed that the set of intervened variables $\mathcal{I}$ does not contain cycles among themselves — that is, the subgraph induced by $\mathcal{I}$ must be acyclic. This condition ensures that there is a unique assignment between unmatched rows and the indices in $\mathcal{I}$ such that the resulting matrix has non-zero diagonal entries. Moreover, it can be verified directly from the observational weight matrix: one can check whether the corresponding submatrix admits a unique row permutation that yields non-zero diagonal entries. If so, we can safely intervene on the entire set $\mathcal{I}$ at once and recover the true corresponding row for each variable, assigning it to its correct index.
>
> * Topic 5: Sampling-Based Estimation and Optimality -
> This question fits within the adaptive submodularity framework of Golovin & Krause (2011), which offers approximation guarantees for greedy policies even with approximate marginal gains. They showed that if each selection has an *absolute error* of at most $\varepsilon$, the total reward remains near-optimal, with only an additive degradation.
> Our greedy rule selects the intervention $v$ maximizing the (estimated) marginal gain
> $\Delta(v\mid\psi)=N\,L(p_v)$ with $L(p_v)=\sum_{i=1}^{k_v} p_{v,i}(1-p_{v,i})$, where $p_v$ is the categorical distribution over candidate rows adjacent to $v$ in the bipartite graph. We estimate $p_v$ from $M$ i.i.d. samples and use $ L(q_v) $ as our estimation for $L(p_v)$, where $ q_v $ is the empirical distribution formed from $ M $ i.i.d. samples used to estimate $p_v$. Theorem 7.1 in our paper shows that, for any fixed $v$,
> $$\mathbb{P}\left[\bigl|L(q_v)-L(p_v)\bigr|\ge \tfrac{1}{M}+\sqrt{\tfrac{2}{M}\log\left(\tfrac{2}{\delta}\right)}\right]\le\delta,\tag{1} \quad\text{with prob. } \ge 1-\delta.$$
> hence
> $$
> \bigl|\widehat{\Delta}(v\mid\psi)-\Delta(v\mid\psi)\bigr|
> \le
> N\left(\tfrac{1}{M}+\sqrt{\tfrac{2}{M}\log\left(\tfrac{2}{\delta}\right)}\right)
> \quad\text{with prob. } \ge 1-\delta.$$
> where $\widehat{\Delta}(v\mid\psi) = NL(q_v)$. Now we want to have **uniform control across candidates.**
> Let $A_\psi$ be the set of feasible interventions at state $\psi$ (typically $|A_\psi|\le n$). Applying a union bound to eq. (1) with $\delta'=\delta/|A_\psi|$ yields, with probability at least $1-\delta$,
> $$\sup_{v\in A_\psi}\bigl|\widehat{\Delta}(v\mid\psi)-\Delta(v\mid\psi)\bigr|
> \le
> R_M(\delta,|A_\psi|)
> :=
> N\left(\tfrac{1}{M}+\sqrt{\tfrac{2}{M}\log\Bigl(\tfrac{2|A_\psi|}{\delta}\Bigr)}\right).\tag{2}$$
> Let $v^\star\in\arg\max_v \Delta(v\mid\psi)$ and $v_{\text{sel}}\in\arg\max_v \widehat{\Delta}(v\mid\psi)$.
> By optimality of $v_{\text{sel}}$ for the estimates and the two–sided deviation bound in eq. (2), with probability at least $1-\delta$, we have
> $$\Delta(v_{\text{sel}}\mid\psi)
> \ge
> \widehat{\Delta}(v_{\text{sel}}\mid\psi)-R_M
> \ge
> \widehat{\Delta}(v^\star\mid\psi)-R_M
> \ge
> \Delta(v^\star\mid\psi)-2R_M .\tag{3}$$
> Thus each greedy step is computed with **absolute error** at most
> $\varepsilon_M:=2R_M(\delta,|A_\psi|)$.
> Since our objective is adaptive monotone and adaptive submodular, we can invoke the **guarantees of Golovin and Krause (2011) for the greedy rules implemented with only a small absolute error**. Please refer to this paper for the definition of Policy Truncation (Definition 4), and Theorem 5. Hence, if at every step the selected item satisfies eq. (3), then for any integers $\ell,k>0$,
> $$F\left(\pi_{[\ell]}\right) \ge \left(1-e^{-\ell/k}\right)F\left(\pi_{[k]}^{\star}\right) - \ell\varepsilon_M,$$
> with high probability.
> Therefore, the cumulative degradation scales as
> $O\left(\ell N\sqrt{\tfrac{\log(|A_\psi|/\delta)}{M}}\right)$, vanishing at the standard Monte-Carlo rate.
>
> * Topic 6: Number of samples ($M$) used in the sampling-based estimation -
> We experimented with $M \in \lbrace100, 500, 1000, 3000 \rbrace$ and found that $M = 1000$ offered a good balance between accuracy and computational cost. Smaller values led to noisy estimates and worse interventions, while larger ones gave marginal gains with high runtime. Thus, we fixed $M = 1000$ in all experiments and will note this in the revision.
>
> Pearl, Linear Models: A Useful Microscope for Causal Analysis, Journal of Causal Inference, 2013.
>
> Shimizu, LiNGAM: Non-Gaussian Methods for Estimating Causal Structures." Behaviormetrika, 2014.
>
> Hoyer et al., Estimation of Causal Effects using Linear non-Gaussian Causal Models with Hidden Variables. International Journal of Approximate Reasoning, 2008.
>
> Salehkaleybar et al., Learning Linear non-Gaussian Causal Models in the Presence of Latent Variables, JMLR 2020.
>
> Kivva et al., A Cross-Moment Approach for Causal Effect Estimation, NeurIPS 2023.
>
> Peters et al., Elements of Causal Inference: Foundations and Learning Algorithms, 2017.
>
> Åström and Murray, Feedback Systems: An Introduction for Scientists and Engineers, 2008.
>
> Ljung, System Identification: Theory for the User, 1999.
>
> Eriksson and Koivunen, Identifiability, Separability, and Uniqueness of Linear ICA Models." IEEE Signal Processing Letters, 2004.
> Shimizu et al., A Linear non-Gaussian Acyclic Model for Causal Discovery, JMLR, 2006.
>
> Golovin, Daniel, and Andreas Krause. "Adaptive submodularity: Theory and applications in active learning and stochastic optimization." Journal of Artificial Intelligence Research 42 (2011)

---

> > ### Comment · Reviewer_kRFZ · 2025-08-02
> > **Re.**
> >
> > I thank the reviewer for the detailed response to my questions. Including some discussion from the response will improve the manuscript. I will keep score for the pape

---

### Official Review · Reviewer_k8AX · 2025-07-01

**Clarity:** 4
**Significance:** 3
**Originality:** 4
**Rating:** 5
**Confidence:** 4

**Summary:**

This paper tackles causal structure learning in linear non-Gaussian models with cycles, where observational data alone only identifies graphs up to an equivalence class. The authors provide a novel combinatorial characterization showing that each equivalence class corresponds to a perfect matching in a bipartite graph, and prove that each intervention reveals one edge of the true matching while eliminating incompatible graphs. They formalize optimal experiment design as an adaptive stochastic optimization problem with a submodular reward function, enabling a greedy algorithm with near-optimal performance guarantees. To make this computationally tractable, they develop a sampling-based estimator that avoids explicitly enumerating all graphs in the equivalence class and provide theoretical analysis of its accuracy. Their experiments in synthetic linear non-Gaussian data demonstrate that, when provided with the oracle weight matrix up to row-permutation and scaling, this approach recovers true causal structures with few interventions, often performing close to the theoretical lower bound.

**Questions:**

Q1. How does your interventional design approach perform when the initial observational equivalence class is incorrectly estimated? To address this, please consider doing the following:
   - Provide theoretical analysis showing consistency can still be achieved even with incorrect initial equivalence class estimation (acknowledging optimality may be lost)
   - Develop a modified implementation that accounts for uncertainty in the equivalence class (e.g., by sampling from nearby equivalence classes)
   - Present empirical results on synthetic data quantifying how sensitive your approach is to misspecification

Q2. Which components of your approach could potentially extend to nonlinear causal models, and where does linearity play a fundamental role? To address this, please consider doing the following:
   - Clearly identify which theoretical results (bipartite matching characterization, strongly connected components analysis, intervention selection) rely essentially on linearity vs. those that might generalize
   - Discuss whether the strongly connected components framework could be useful for characterizing equivalence classes in nonlinear settings
   - Provide informed speculation about what modifications would be needed for nonlinear extensions

If the authors address my concerns in the Strengths/Weaknesses and Questions sections, I am very happy to increase my score.

**Ethical Concerns:**

["NO or VERY MINOR ethics concerns only"]

**Final Justification:**

The paper develops a novel theoretical approach to an interventional structure learning in graphs with cycles, an important and often unaddressed problem in causal discovery. It also presents interesting causal sub-structures such as the SCC and other proof techniques that may be generalizable beyond the linear setting considered. The authors provided additional experimental results that confirm the robustness of their approach when the initial MEC estimated gleaned from observational data has errors. I recommend to accept.

**Limitations:**

Limitations, outside of the additional empirical experiments I requested above, are adequately addressed.

**Paper Formatting Concerns:**

None.

**Quality:**

4

**Strengths And Weaknesses:**

Strengths:

Clarity - This submission is very clearly written, and extremely well organized. Assumptions are clearly stated in assumption formatting in Section 3, all reocurring terminology is defined and explained, and the groundwork for upcoming sections. The major sections are sensibily split into mapping each observational equivalence class into a bipartite graph (4), characterizing how interventions pair down the equivalence class (5), developing an adaptive greedy interventional design that is provably near optimal (6), and suggesting a reasonable statistical estimator for the theoretical method in 6, with corresponding analysis of estimator error. In each section, the authors very seamlessly weave between definitions and mathematical results, and explaining the intuition behind each step.

Quality - Although I did not check the proofs in the appendix closely, the intuition behind the major results (Theorem 4.4, Theorem 6.6, and Theorem 7.1) seems clear, and proof sketches throughout the work seem accurate. The authors claims seem generally reasonable, as they outline weaknesses (such as the computational intractability of the theoretical reward function in Section 6), and take steps to try and mitigate them (developing a statistical estimator via sampling in Section 7). The authors adress a wide range of parts of the discovery problem, providing a novel framing of equivalence classes as bipartite matchings, connecting interventional results to the matching, and finally prove near statistical optimality of their approach. Therefore, the claimed contributions in Section 1 are largely substantiated by the details of the work.

Originality - The key novelty in the work lies first in finding a bipartite matching representation of the equivalence class of potentially cyclic linear SEMs (Section 4), then showing how this bipartite matching can be combined with interventions to eliminate graphs from the class (Section 5). The most important idea is perhaps the move in Section 4.1 to reduce the typical $d$-dimensional graph to a lower level representation  (condensation graph) via strongly connected components, and then proving that condensation graphs are sufficient for acharacterizing all distribution-equivalent graphs. This analysis is quite neat, and potentially more broadly applicable beyond linear SEMs, as strongly connected components may be useful (but perhaps not sufficient) for characterizing other distributions.

Weaknesses:

Significance

- The main simplifying assumption int his work is linearity (and also causal sufficiency to some degree), which limit the scope of applicability. While this is fine for foundational results, and particularly as there are not many works addressing interventional experiment design in cyclic settings, authors should make clear where they leverage linearity heavily. Additional a discussion on what parts of this are potentially transferable to the nonlinear setting, and perhaps some speculation on what parts are more or less likely to generalize would also be valuable to other researchers in this area.
- The other major drawback of the experimental approach here is that it seems to depends somewhat strongly on the accuracy of the original estimation of the equivalence class. After all, the way interventions are chosen depends on calculating the value of the  intervention subject to knowledge of the class. This motivates the need to characterize how their approach interacts with error in estimating the original observational class. This can be tackled theoretically in different ways; either you could show that consistency can still be achieved even if original test was wrong (optimality is probably no longer true though), or you could provide a way to loosen the current implementation to get consistency, perhaps by randomly sampling from graphs in nearby equivalence classes (just a thought). This could also be addressed empirically, by running tests in synthetic data to see how bad a problem misspecification of the original equivalence class is - this would help inform future work as to whether this is an subarea of this methodology that would benefit a lot from improvement.

---

> ### Author Rebuttal · Authors · 2025-07-31
>
> * Topic 1: Errors in Estimating the Equivalence Classes
>     * Analysis of how our method behaves under misspecification -
> When the initial observational equivalence class is incorrectly estimated (e.g., due to ICA errors), the algorithm may still make progress by eliminating incorrect matchings that conflict with interventional data. However, theoretical guarantees such as consistency or optimality can no longer be ensured in the worst case, as the algorithm may converge to an incorrect row permutation of the ground-truth matrix according to our extended simulations (see the results in the following). Still, in most cases, the algorithm recovers the true structure.
>     * Algorithmic modifications to deal with uncertainty in the equivalence class -
>      We have extended our experiments to a more realistic setting that does not rely on access to an oracle ICA procedure. Instead, we simulate the practical case where ICA is estimated from finite samples by applying *FastICA* to both the observational and interventional datasets, which may lead to an uncertainty in the equivalence classes. To ensure robustness in the presence of ICA estimation noise and corresponding misspecifications in the equivalence class, we made two key algorithmic modifications:
>         * **Adaptive Thresholding:** After ICA recovery, the resulting rows may be noisy, requiring the elimination of small-magnitude elements. However, naive thresholding can lead to overly sparse graphs that no longer admit a perfect matching in the corresponding bipartite representation. To address this, we implemented an adaptive thresholding scheme: it begins with strict relative and absolute magnitude thresholds and progressively relaxes them until the resulting bipartite graph admits at least one perfect matching.
>         * **Safe Matching via Ranked Edge Removal:** To avoid incorrect row matches (which could result in not finding perfect matchings), we replaced the original best-match strategy with a conservative method. For each unmatched node, we rank candidate rows by similarity and attempt to remove each from the bipartite graph. A candidate is only accepted if the resulting bipartite graph still admits a perfect matching.
>     * Empirical evidence showing robustness against misspecifications -
>     With these modifications, we observe that our method continues to outperform all baseline strategies (e.g., Max Degree and Random) in terms of the number of experiments needed for full causal discovery, even under noisy ICA-based estimation:
>
> **Table**: Average number of experiments required for full causal discovery in different methods (finite-sample ICA setting)
>
> | **Number of Variables** | **FVS** | **Our Method** | **Max Degree** | **Random** |
> |-------------------------|--------:|----------------:|----------------:|-----------:|
> | 4                       |   0.12  |           0.19  |           0.20  |      0.23  |
> | 5                       |   0.28  |           0.34  |           0.40  |      0.39  |
> | 6                       |   0.42  |           0.44  |           0.56  |      0.63  |
> | 7                       |   0.54  |           0.62  |           0.75  |      0.91  |
> | 8                       |   0.82  |           0.89  |           1.15  |      1.26  |
> | 9                       |   0.88  |           1.11  |           1.52  |      1.65  |
> | 10                      |   1.11  |           1.35  |           1.91  |      2.33  |
>
> We also report the distribution of relative errors
> $\varepsilon_{\mathrm{rel}}=\frac{\Vert \widehat{W}-W^* \Vert_{F}}{\Vert W^* \Vert_{F}}$ between the ground-truth $W^*$ and recovered matrix $\widehat{W}$ across all runs. In over 80\% of runs, the error is less than 15\%, suggesting that—even under misspecification of the initial equivalence class—our approach retains strong empirical performance, although the guarantee of consistency may no longer hold:
>
> **Table**: Distribution of Relative Errors Across All Runs
>
> | **Relative Error Range** | **Number of Instances** |
> |--------------------------|-------------------------:|
> | < 0.05                   |                      501 |
> | 0.05 – 0.15              |                       58 |
> | 0.15 – 0.25              |                       17 |
> | > 0.25                   |                      124 |
>
>
>
> * Topic 2: Extentions to Nonlinear Settings
>     * Justification of linearity assumption
>         Linear models are prevalent in many applications, mainly because they can be learned with moderate sample size and offer simple qualitative interpretations. On the theory side, understanding causal phenomena in a linear model is often a necessary step toward a complete understanding of the given phenomenon, see e.g., (Pearl, 2013). Moreover, in many physical and engineering systems, linear models provide accurate approximations around equilibrium points or typical operating regimes. This local linearity is a well-established principle in control theory and system identification (Åström and Murray, 2008; Ljung, 1999), and supports the use of linear assumptions as a starting point even when the underlying system may be globally nonlinear. The non-Gaussianity assumption is a common assumption required in a wide range of literature on causal effect estimation and causal discovery (Shimizu 2014, Hoyer et al. 2008, Salehkaleybar et al. 2020, Kivva et al. 2023, Peters et al. 2017). More specifically, for Gaussian distributions, some causal relationships are indistinguishable from observational data. This issue is well-illustrated by the non-identifiability results for linear Gaussian models from observational data (Peters et al. 2017).
>     * Generalization of our results to nonlinear settings
>     In general SCMs, with the set of structural equations, $X_i:=f_i(Pa_i,N_i), 1\leq i\leq n$, let $\mathbf{f}$ be a vector-valued function where the $i$-th coordinate is $f_i$. In (Reizinger et al., 2023), for DAGs, it is shown that the support of Jacobian of $\mathbf{f}^{-1}$ is equal to $I-A$ where $A$ is the adjacency matrix of the causal graph. We believe that their proof can be extended to general directed graphs under the same conditions as in Assumption 1 in their work, excluding part (i), which is about acyclicity. Moreover, several algorithms in non-linear ICA have been proposed which can learn $\mathbf{f}^{-1}$ up to some scaling, permutation, and monotonic element-wise transformations such as (Khemakhem et al. 2020) or (Kivva et al. 2022). Therefore, these results can be utilized to learn the $\mathbf{f}^{-1}$ up to the mentioned indeterminacies and then used to learn the rows of $I-A$ up to some scaling and permutation. Let us denote the recovered matrix by $I - A_{\text{ICA}}$. To recover SCCs, we can utilize the recovered matrix $I - A_{\text{ICA}}$, which allows us to identify the equivalence class obtained from the observational data.
> For the experiment design, which incorporates interventional data, we now rely on the matrix $I - A$ (as opposed to $I - W_{\text{ICA}}$ in the linear case). This approach requires that the sparsity patterns of the rows be distinct to enable correct matching in the bipartite graph. Consequently, based on current results in nonlinear ICA, it is needed that each node has a unique parent set in order to carry out the experiment design step. We hope that this assumption can be relaxed with future advances in nonlinear ICA.
>
> Pearl, Linear Models: A Useful Microscope for Causal Analysis, Journal of Causal Inference, 2013.
>
> Shimizu, LiNGAM: Non-Gaussian Methods for Estimating Causal Structures." Behaviormetrika, 2014.
>
> Hoyer et al., Estimation of Causal Effects using Linear non-Gaussian Causal Models with Hidden Variables. International Journal of Approximate Reasoning, 2008.
>
> Salehkaleybar et al., Learning Linear non-Gaussian Causal Models in the Presence of Latent Variables, JMLR 2020.
>
> Kivva et al., A Cross-Moment Approach for Causal Effect Estimation, NeurIPS 2023.
>
> Peters et al., Elements of Causal Inference: Foundations and Learning Algorithms, 2017.
>
> Åström and Murray, Feedback Systems: An Introduction for Scientists and Engineers, 2008.
>
> Ljung, System Identification: Theory for the User, 1999.
>
>  Khemakhem et al., Variational Autoencoders and Nonlinear ICA: A Unifying Framework, AISTATS 2020.
>
> Kivva et al., Identifiability of Deep Generative Models without Auxiliary Information, NeurIPS 2022.
>
> Reizinger et al., Jacobian-based Causal Discovery with Nonlinear ICA, Transactions on Machine Learning Research, 2023.

---

> > ### Comment · Reviewer_k8AX · 2025-08-04
> >
> > I thank the authors for their thoughtful response and for addressing the points I raised. I will maintain my positive score, and am inclined to raise it further if the authors agree to incorporate the additional experimental results from 'safe matching via ranked edge removal' (Topic 1) and to include a discussion on the potential extension to the nonlinear setting (Topic 2) in the camera-ready version.

---

> > > ### Author Response · Authors · 2025-08-04
> > >
> > > We thank the reviewer for the valuable comments. In the final version, we will include experimental results for "safe matching" and add a discussion on how the current work can be extended to nonlinear settings.

---

### Official Review · Reviewer_1ET8 · 2025-07-03

**Clarity:** 3
**Significance:** 3
**Originality:** 4
**Rating:** 5
**Confidence:** 3

**Summary:**

This work addresses interventional experiment design in cyclic causal models under linear non-Gaussian functional models.
It contributes a characterization of the (observational) Markov Equivalence class in this setting through a direct correspondence between graphs in the MEC and matchings
in bipartite graphs. The authors give an analysis of how targeted interventions on the system constrain this class, and motivate a reward function for optimal experiment design
for which they establish near-optimality guarantees. A sampling-based estimator is also suggested and analyzed, and simulation experiments support both the exact and sampling-based strategies.

**Questions:**

The main questions are related to the concerns mentioned above,
- How does the method perform against previous proposals for optimal experimental design? Even when these assume acyclicity, it would be nevertheless be interesting to see the empirical performance in a cyclic setting. Similarly, it would be interesting to see a comparison between the acyclic vs. cyclic setting to judge the effect of this.
- Did you perform any experiments without access to an ideal ICA procedure to judge the effect of this, and if not do you expect the version without ICA-oracle to still outperform the baselines?

**Ethical Concerns:**

["NO or VERY MINOR ethics concerns only"]

**Final Justification:**

The detailed rebuttal answers adequately address all points raised by the reviewers. The main points that were raised and resolved in the rebuttal and discussion are

- addressing the assumption of access to an ideal ICA procedure through an additional experiment;
- resolving a concern related to sampling-based estimation using results of Golovin & Krause (2011);
- providing a detailed argument showing uniformity of the posterior under the uniform-prior assumption; and
- giving additional clarifications and justifications, such as justifying the assumptions on the data-generating process, discussing extensions for non-linear settings, and clarifying that the approach allows for multi-node interventions.

As long as the above points meaningfully integrated into the revised manuscript, I am willing to vote for acceptance.

**Limitations:**

There a no limitations besides what is noted above and no negative impacts.

**Paper Formatting Concerns:**

There are no major formatting issues.

**Quality:**

3

**Strengths And Weaknesses:**

**Strengths** The problem setting is sufficiently novel compared to the existing work  (e.g. Mokhtarian, 2023)  making this work a valuable addition to the literature.
In particular, the theoretical insights are original, both regarding the graphical characterizations as well as the proposed adaptive greedy policy with guarantees and sampling-based estimator.
Theoretical justifications of all of these aspects are given where I found no major issues. There are also no issues regarding clarity and presentation of the work.

**Weaknesses**  Regarding the quality of the experiments,
it would be interesting  to see a more comprehensive experimental comparison to adaptive methods assuming acyclicity, perhaps on both  cyclic and acyclic settings, to better judge the empirical benefits of considering this setting.
Experiments are conducted assuming access to an optimal ICA strategy which, while  reasonable  to isolate the effect of the proposed strategy itself, leave it unclear whether the proposal allows to address real-world settings.
This affects the practical significance of this work, which may be somewhat limited by to the strong assumptions on the generating process,
namely the linear non-Gaussian model and perfect, single-node interventions without off-target effects  which may not always be realistic.

Nevertheless, existing work addressing cyclic causal graphs is indeed limited and this work can provide an important basis for future work, in particular with the given characterization of the MEC which may be of independent interest.

---

> ### Author Rebuttal · Authors · 2025-07-31
>
> * Topic 1: Robustness to Non-Ideal ICA - Finite Sample:
> We have extended our experiments to a more realistic setting that does not rely on access to an oracle ICA procedure. Instead, we simulate the practical case where ICA is estimated from finite samples by applying *FastICA* to both the observational and interventional datasets. To ensure the robustness of the matching procedure under finite-sample noise, we made a few modifications to the algorithm:
>     * **Adaptive Thresholding:** After ICA recovery, the resulting rows may be noisy, requiring the elimination of small-magnitude elements. However, naive thresholding can lead to overly sparse graphs that no longer admit a perfect matching in the corresponding bipartite representation. To address this, we implemented an adaptive thresholding scheme: it begins with strict relative and absolute magnitude thresholds and progressively relaxes them until the resulting bipartite graph admits at least one perfect matching.
>     * **Safe Matching via Ranked Edge Removal:** To avoid incorrect row matches (which could result in not finding perfect matchings), we replaced the original best-match strategy with a conservative method. For each unmatched node, we rank candidate rows by similarity and attempt to remove each from the bipartite graph. A candidate is only accepted if the resulting bipartite graph still admits a perfect matching.
> With these modifications, we observed that our method continues to outperform all baseline strategies (e.g., Max Degree and Random), as shown in the table below:
>
> **Table**: Average number of experiments required for full causal discovery in different methods (finite-sample ICA setting)
>
> | **Number of Variables** | **FVS** | **Our Method** | **Max Degree** | **Random** |
> |-------------------------|--------:|----------------:|----------------:|-----------:|
> | 4                       |   0.12  |           0.19  |           0.20  |      0.23  |
> | 5                       |   0.28  |           0.34  |           0.40  |      0.39  |
> | 6                       |   0.42  |           0.44  |           0.56  |      0.63  |
> | 7                       |   0.54  |           0.62  |           0.75  |      0.91  |
> | 8                       |   0.82  |           0.89  |           1.15  |      1.26  |
> | 9                       |   0.88  |           1.11  |           1.52  |      1.65  |
> | 10                      |   1.11  |           1.35  |           1.91  |      2.33  |
>
> We also report the distribution of relative errors $\varepsilon_{\mathrm{rel}}  = \frac{\Vert \widehat{W} - W^* \Vert_{F}}{\Vert W^* \Vert_F}$ between the ground-truth $W^*$ and recovered matrix $\widehat{W}$ across all runs.
> In over 80\% of cases, the error is less than 15\%, which confirms that our conservative matching design ensures reliable recovery despite finite-sample noise:
>
> **Table**: Distribution of Relative Errors Across All Runs
>
> | **Relative Error Range** | **Number of Instances** |
> |--------------------------|-------------------------:|
> | < 0.05                   |                      501 |
> | 0.05 – 0.15              |                       58 |
> | 0.15 – 0.25              |                       17 |
> | > 0.25                   |                      124 |
>
> * Topic 2: Assumptions in the Data Generating Process:
>     * Justification of linear non-Gaussian model -
>     Linear models are prevalent in many applications, mainly because they can be learned with a moderate sample size and offer simple qualitative interpretations. On the theory side, understanding causal phenomena in a linear model is often a necessary step toward a complete understanding of the given phenomenon, see e.g., (Pearl, 2013). Moreover, in many physical and engineering systems, linear models provide accurate approximations around equilibrium points or typical operating regimes. This local linearity is a well-established principle in control theory and system identification (Åström and Murray, 2008; Ljung, 1999), and supports the use of linear assumptions as a starting point even when the underlying system may be globally nonlinear.
> The non-Gaussianity assumption is a common assumption required in a wide range of literature on causal effect estimation and causal discovery (Shimizu 2014, Hoyer et al. 2008, Salehkaleybar et al. 2020, Kivva et al. 2023, Peters et al. 2017). More specifically, for Gaussian distributions, some causal relationships are indistinguishable from observational data. This issue is well-illustrated by the non-identifiability results for linear Gaussian models from observational data (Peters et al. 2017).
>     * Generalization to multi-node interventions -
>     Our method is not restricted to single-node interventions. We briefly outline the extension.
>     When an intervention is applied to a set $\mathcal{I} = \lbrace i_1, i_2, \dots, i_t \rbrace$, the post-intervention matrix differs from the observational one only in the $t$ rows corresponding to the intervened variables. By comparing these matrices after ICA, we can identify the unmatched rows and associate each with an element of $\mathcal{I}$, as long as the variables in $\mathcal{I}$ do not form cycles among themselves — that is, the subgraph induced by $\mathcal{I}$ is acyclic.
> This condition ensures that there is a unique assignment between unmatched rows and the indices in $\mathcal{I}$ such that the resulting matrix has non-zero diagonal entries. Moreover, it can be verified directly from the observational weight matrix: one can check whether the corresponding submatrix admits a unique row permutation that yields non-zero diagonal entries. If so, we can safely intervene on the entire set $\mathcal{I}$ at once and recover the true corresponding row for each variable, assigning it to its correct index. This enables safe parallelization of interventions by selecting such subsets, thereby leading to more efficient experimental design.
>     * Generalization to imperfect interventions -
>     Our approach also extends to imperfect interventions, provided they sufficiently perturb the variable such that its post-intervention row does not match any row from the observational data. This condition holds with high probability under many realistic intervention mechanisms, such as random modifications of parental relationships. Therefore, our method remains valid and informative even under imperfect or noisy interventions.
> * Topic 3: Comparison to Adaptive Methods for Acyclic Graphs
> A summary of existing work on adaptive experiment design in the acyclic setting is provided in Table 1 of (Elahi et al., 2024). Most of the existing methods conduct experiments in the infinite-sample regime—that is, assuming access to the exact observational/interventional Markov equivalence class (MEC). However, since the equivalence classes defined for acyclic graphs do not extend to the cyclic case, these algorithms cannot be directly applied in the infinite-sample regime for cyclic settings. To the best of our knowledge, Elahi et al. (2024) is the only recent work that has conducted experiments in the finite-sample regime. Unfortunately, their implementation is not publicly available, and due to the short rebuttal period, we were unable to reimplement their method. Nonetheless, we will implement their method and report the results in the final version. That said, we believe that adaptive algorithms designed specifically for the acyclic setting may perform poorly in cyclic settings, as they do not account for the presence of cycles and often rely on conditional independence tests, which may not be sample-efficient.
> Pearl, Linear Models: A Useful Microscope for Causal Analysis, Journal of Causal Inference, 2013.
>
> Shimizu, LiNGAM: Non-Gaussian Methods for Estimating Causal Structures." Behaviormetrika, 2014.
>
> Hoyer et al., Estimation of Causal Effects using Linear non-Gaussian Causal Models with Hidden Variables. International Journal of Approximate Reasoning, 2008.
>
> Salehkaleybar et al., Learning Linear non-Gaussian Causal Models in the Presence of Latent Variables, JMLR 2020.
>
> Kivva et al., A Cross-Moment Approach for Causal Effect Estimation, NeurIPS 2023.
>
> Peters et al., Elements of Causal Inference: Foundations and Learning Algorithms, 2017.
>
> Åström and Murray, Feedback Systems: An Introduction for Scientists and Engineers, 2008.
>
> Ljung, System Identification: Theory for the User, 1999.
>
> Elahi et al., Adaptive Online Experimental Design for Causal Discovery, arXiv preprint arXiv:2405.11548, 2024.

---

> > ### Comment · Reviewer_1ET8 · 2025-08-04
> >
> > **Topic 1** Thank you for running experiments with a nonideal ICA procedure, it is good to see that the approach is robust to this.
> >
> > **Topic 2**  The comments on how the multi-node and imperfect interventions can be addressed would be helpful to include in the paper if space allows. Given that such interventions can be accommodated and that the linear non-Gaussian assumption is indeed a reasonable starting point in many applications, I consider the overall assumption set of this work reasonable.
> >
> > **Topic 3** The efforts in re-implementing the approach by Elahi et al. (2024) are much appreciated, and it is fully understandable that this is not possible within the short rebuttal period. It is indeed reasonable that methods assuming acyclicity likely perform poorly in this setting, but it will be good to see at least one such comparison in the experiments.
> >
> > Given the current state and the adequately addressed questions, I am willing to increase my rating if no concerns remain after the feedback/discussion from the other reviewers.

---

> > > ### Author Response · Authors · 2025-08-04
> > >
> > > We thank the reviewer for the valuable comments. In the revised manuscript, we will include a discussion regarding multi-node and imperfect interventions. Additionally, for completeness, we will add a comparison with experimental design algorithms specifically developed for DAGs.

---

### Decision · Program_Chairs · 2025-09-17

**Decision:**

Accept (spotlight)

**Comment:**

The paper focuses on intervention design in cyclic linear non-Gaussian models. It proposes a greedy policy for the interventions with near-optimal guarantees and a sampling-based estimator to efficiently estimate the reward function without enumerating all graphs in the (I)MEC. The reviewers acknowledge that the paper proposes a novel and well-founded theoretical analysis and method, mentioning also that the paper is very clearly written and rigorous. On the other hand, the reviewers raised a few minor issues regarding using more ablation and comparisons in the experimental evaluation, and regarding the significance, given its constraining parametric assumptions and single-node interventions. The authors provided several ablations and additional experimental results in the rebuttal, as well as a sketch of a generalization to imperfect and multi-node interventions, thus addressing in detail all of the reviewers' concerns. Overall, I agree with the reviewers that this is an interesting and theoretically sound contribution to the intervention design literature and hence recommend the paper's acceptance.